# Rudimentary Assessment of Waste-to-Wealth of Used Tires Crumbs in Thermal Energy Storage

Hussain H. Al-Kayiem [1,*], Bilawal A. Bhayo [2], Elena Magaril [3] and Pavithra Ravi [4]

1 Mechanical Engineering Department, Universiti Teknologi PETRONAS, Seri Iskandar 32610, Malaysia
2 Mechanical Engineering Department, Mehran University of Engineering & Technology, Jamshoro 76062, Pakistan; bilawal.ahmed@muetkhp.edu.pk
3 Department of Environmental Economics, Ural Federal University, 19 Mira Str., Office I-418A, 620002 Ekaterinburg, Russia; magaril67@mail.ru
4 PETRONAS, Kerteh 24300, Malaysia; ravipavithra1912@gmail.com
* Correspondence: hussain_kayiem@utp.edu.my

**Abstract:** Disposing of waste tires is a major environmental and economic issue. Different recycling methods have been studied to account for its re-usage. This project aims to evaluate the possible usage of shredded waste tires in thermal energy storage (TES) applications, whether they are sensible or latent materials. An experimental setup has been developed with seven compartments. Each compartment contains different TES materials, including tire crumbs, paraffin wax, paraffin wax with shredded tires, pebbles, pebbles with shredded tires, concrete, and concrete with shredded tires. In all cases of the mixture, the base materials are 60%*vol*, and the tire crumbs are 40%*vol*. The experimental included three locations for temperature measurements in each compartment, solar irradiation, and ambient temperature. The tests were carried out from 9:00 a.m. till 7:00 p.m. and repeated for five days to account for the weather's daily change. Results revealed that mixed 60%*vol* pebbles and 40%*vol* shredded tires have the highest recorded temperature, at 112.5 °C, with a 39.5% increment compared to pure pebbles. The interesting finding is that the added tire crumbs reduced the storage capacity of the paraffin wax, which is latent TES material. At the same time, it increased the storage capacity of the concrete and pebbles, which are sensible TES materials. Adding 40%*vol* of tire crumbs to the paraffin wax has a negative effect, where the thermal storage capacity is reduced by 43%, and the discharge capacity is reduced by 57%. In contrast, the concrete and the pebbles show enhanced storage capacity. Adding 40%*vol* of crumbs to the concrete increased the charging capacity by 54% and discharging capacity by 33.7%. The 40%*vol* added tire crumbs to the pebbles increased its charging capacity by 25% and the discharging capacity by 33%. The rudimentary assessment encourages further investigations on using the wasted tires crumbs for TES. The results reveal the probability of a circular economy using wasted tires with sensible TES for solar-to-thermal energy conversion.

**Keywords:** circular economy; latent TES; recycling of waste tires; sensible TES; solar thermal energy; TES; tire crumbs; storage capacity



## 1. Introduction

Waste tires recycling is gaining new soundness in the transition to a circular economy, which has become a development priority in many countries [1–3]. Currently, more than 250 million new tires are produced every year. After being used for automobiles, trucks, buses, aircraft landing gear, tractors, and other farm equipment, industrial vehicles such as forklifts, and common conveyances such as baby carriages, shopping carts, wheelchairs, bicycles, and motorcycles, have become a major global waste problem. The introduction and relevant literature of the current article are subdivided into four subsections.

### 1.1. What Are Tires

Collectively, around 1.5 billion tires reach the end of their useful lives every year. They can be retread and reused up to a point, but not endlessly. End-of-life tires (ELTs) are difficult to process for recycling because of their complex mix of materials. A tire is a strong, flexible rubber casing attached to the rim of a wheel. They are made of a composite of different materials. Generally, tires are composed of 14% natural rubber; 27% synthetic rubber produced from unsaturated hydrocarbons; 16% of fabrics and fillers; 14% of carbon steel wires for reinforcement; and 28% of carbon steel black [4,5]. Figure 1 shows the typical composition of the tires.

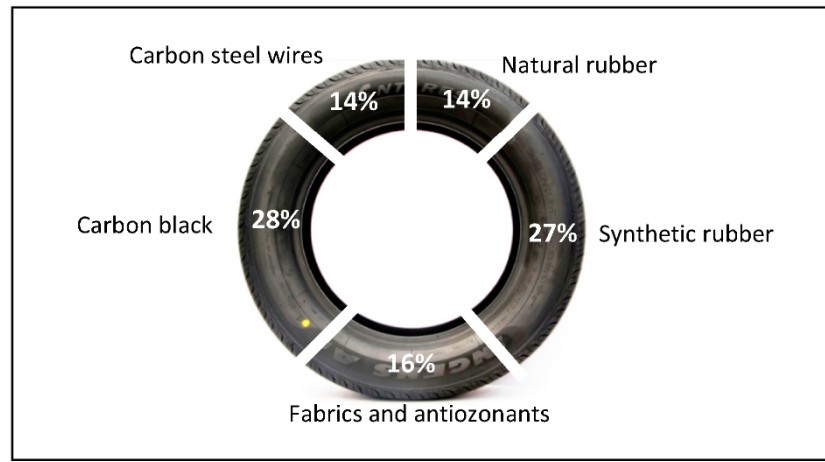

**Figure 1.** Typical composition of the tire.

### 1.2. Tires as Waste

The disposal of waste tires has become a global major environmental concern due to the increase in automobile usage, especially in highly industrialized areas with large populations. The problems caused by the waste tires are major because they are not biodegradable and can last for several decades if no proper handling is carried out [6].

Car tires, which are not biodegradable, are a big global waste concern. Approximately 1.5 billion tires are derived annually to the end of their useful lives [7,8]. They can be retread and reused to a point, but not permanently, and there is a conundrum at the end of their usefulness. ELTs are difficult to process for any recycling since they are a complicated mixture of materials—natural and synthetic rubber, fiber, and wire, all in a heavy and unwieldy set.

Globally, it is estimated that 13.5 million tons of tires are scrapped annually, 40% of which come from emerging markets such as China, India, South America, Southeast Asia, South Africa, and Eastern Europe [9].

### 1.3. Utilization of Wasted Tires

Rubber recovered from waste tires features a sort of uses. Structural engineering and construction are used for playground surfaces, parking lots, bank stabilization, paved surface filling, and asphalt adjustment [10,11]. Tires have significant building properties, such as lightweight, low ground pressure, good thermal insulation, and good drainage properties [12,13]. Another valuable property is its enhanced damping property, which is sweet for vehicle driving. However, recent fires have stalled the usage of field scrap rubber for several of those applications. In most applications, scrap tires are wont to replace building materials [14,15].

Rubber improved asphalt has increased longevity, decreased reflective cracking, thinner lift, and increased skid resistance [16,17]. Asphalt-modified rubber is additionally used for waterproofing membranes, cracking and joint sealers, hot mixing binders, and roofing materials [6]. The rubber increases the ductility of the asphalt, increasing the temperature at which the asphalt softens. The combination adhesive bond becomes stronger and increases

asphalt time. In building environment applications, rubber is employed for retaining walls, erosion control, barricading of shoring embankments, road embankment fills, and thermal insulation [18].

Wang et al. [19] proposed a model to predict the supply of waste tires in 2019–2023 and the demand for these tires in three road construction scenarios. Furthermore, they evaluated the carbon emissions in the production process of crumb rubber, modified asphalt, and styrene-butadiene-styrene modified asphalt.

### 1.4. Recycling Statistics of Wasted Tires

Ruwona et al. [20] reviewed material and energy recovery from waste tires and found that more than three billion new tires are manufactured annually. However, only 100 million tires are recycled annually by the recycling industry. The tire is extensively built with many complex processes that make it indestructible and create difficulties in recycling tires. However, leading tire recyclers invest massive amounts of money in new technology and machinery that can help recycle tires for various applications and protect the environment. In addition, the tax variance on selling new tires or disposing of old tires plays a key role in the recycling industry [9]. These policies are pushing tire recyclers to invest in equipment and expand tire recycling facilities and are expected to drive the growth of the tire recycling industry. In addition, the increasing implementation of the crumb rubber created from scrap tires promotes the growth of the tire recycling industry.

In 2016, over 30 percent of the crumb rubber used in sports fields and 25 percent of the crumb rubber used on playground surfaces was projected to disrupt the tire recycling industry significantly. Using rubberized asphalt for pavements also creates a pool of opportunities for tire recyclers. It is expected to accelerate the development of the tire recycling industry soon [21].

### 1.5. Utilization of Recycled Tires

Globally, the main directions for waste tires recovery are the production of tire-derived material (52%) and tire-derived fuel (19%) [21,22]. Williams [23] has initiated many ways to reuse products in tires, and because the EU has forbidden the shipping of tires to landfill sites, a lot of research and development has gone into it. Many new companies have arisen to handle end-of-life tires as a resource. One of the most common is the rubberized surfaces used in playgrounds and running tracks, as reported by [24]. Ground-up rubber is now used frequently in asphalt, with many benefits reported by Li [25]. Roads with rubber have a stronger grip and are 50 percent quieter. The last three times were longer than regular tarmac and did not need much maintenance.

The railways are another possible client for ELT [8,26]. Recycled rubber pads may be fitted under rail or tramway tracks to minimize noise and vibration [27]. Switzerland has been doing this since the 1970s. Green Rail is moving further. An Italian company has built a sleeper made from ELTs, lowering the need for concrete and ballast, and they operate quieter. 35 tons of used tires are used for every kilometer of the track using Green Rail sleepers.

Farmers can use waste tires in agricultural applications as barriers to erosion control [28]. Furthermore, the usage of shredded, crumbed, and granulated tires can be considered: shredded tires can be used as fillers for road, rail, and construction purposes [8]. Finely shredded old tires may also be used as long-lasting mulch (protective cover), which is believed to be non-leachable. Rubber mulches (in several colors) have been awarded innovation awards and are commonly used in gardens, parks, playgrounds, and equestrian arenas, which are discussed by [29]. Rubber mulches are said to be permanent and aesthetically pleasing landscape materials.

Tire chips can be shredded to a specific and standardized scale, making them useful in several ways. Filters for wastewater treatment and constructed wetlands are another innovative application. Since tires can be chipped to be more porous than organic compounds, rocks, and other materials, they also serve as better filter media.

Old tires can be used as an alternative fuel in production, the main ingredient in concrete. Whole tires are commonly introduced into cement kilns by rolling them into the upper end of a preheater kiln or dropping them through a slot midway along with a long-wet kiln. High gas temperatures (1000–1200 °C) cause almost immediate, complete, and smokeless tire combustion. Alternatively, the tires are sliced into 5–10 mm chips, in which form they can be injected into the pre-combustion chamber. Some iron input is needed in cement production, so the iron content of steel-belted tires is beneficial to the process, as described by [29].

Waste tires may be milled to produce powder or granules with a particular configuration using different mechanical, cryogenic, and de-vulcanization techniques [30]. However, de-vulcanization methods are rarely used due to their high operating costs. Waste tires could be used as fuel sources. Tires contain the same energy per unit mass as oil and a little more than gas. Therefore, tires can be used as an effective fuel for industrial processes such as power plants with limited negative environmental impacts compared to coal. Energy from the direct combustion of waste tires could be utilized in metal works, paper mills, tire factories, and on a smaller scale, farms, greenhouses, and sewage treatment plants [30]. Eddie and Laboy-Nieves [31] reported a hypothesis of using recycled tires for power generation in Puerto Rico. His study estimated that scrap tires processed with pyrolysis could supply about 379 MWh annually, where scrap tires have a calorific value of 33 MJ/kg.

### 1.6. Wasted Tires as Thermal Energy Storage

The megatrend of solar energy application needs integration with other resources or storage to fulfill the setback of solar interruption during the night and the cloudy days. Thermal energy storage (TES) is the main method of complimenting solar thermal systems. In large-scale solar thermal utilization, expensive materials for thermal energy storage are not economically feasible. Low-cost materials are preferable. Treatment of used wasted tires for affordable usage is not achieved yet. The harm of this source of pollution is growing continuously. Hence, attempting to convert this waste to wealth is essential to reduce its environmental impact [32].

Black carbon in waste tires can collect solar energy in several ways. Carbon black is blended and pressed with exfoliated graphite. This mixture was pressed onto a small copper disc in a preliminary investigation. The disc was connected to the insulated water reservoir by tubing, which generated a small solar geyser. At its height shortly after midday, the temperature of the water had risen to 45 °C. This result indicated that about 4 kW·h/m$^2$ of solar energy had been gained. With average predicted irradiation of around 6 kW·h/m$^2$, it means an efficiency of around 75% [33].

Pyrolysis is a well-known thermochemical method used to decompose waste while restoring desirable goods. Using pyrolysis, reprocessing waste tires to extract high-energy hydrocarbons is a sustainable waste-to-energy solution [34]. Waste tire pyrolysis provides the ability to manufacture high-value-adding products such as oils, carbons, hydrocarbon gases, and steel cords. However, their high energy requirements are a huge drawback to pyrolysis waste. The heat source for tire pyrolysis is mainly provided by burning fossil fuels (e.g., coal or natural gas) or electrical heating. As a result, pyrolysis is also considered an economically unattractive solid waste management strategy. Another approach to waste tire processing could be coking. The technology is environmentally friendly and does not require external energy; as demonstrated in [35], the industrial-scale implementation is required.

Alternative heating techniques are being studied by Abdallah et al. [36] to minimize reliance on fossil fuels/electric power, including auto-thermal pyrolysis and concentrated solar pyrolysis. In auto-thermal pyrolysis, a portion of the products (e.g., gas products or char) is recycled into the furnace to supply the necessary energy for the pyrolysis process [37]., as shown in Figure 2. This approach, however, reduces the process throughput and can cause emission problems [38].

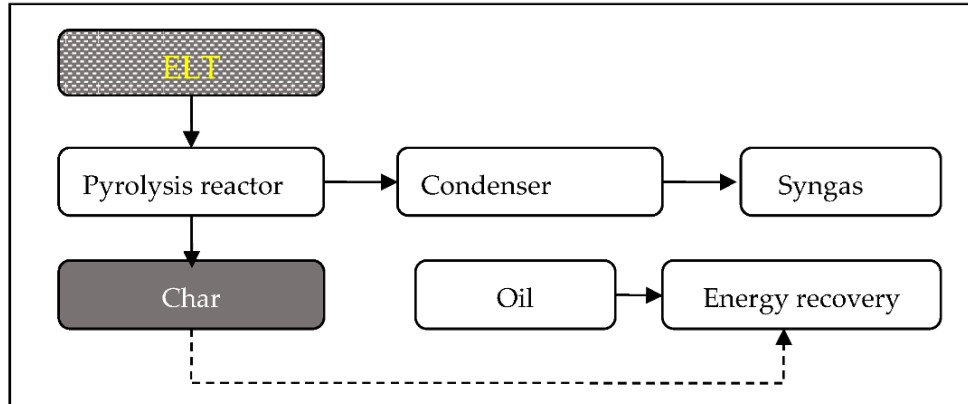

**Figure 2.** The pyrolysis process obtains scrap tires as energy storage.

A careful study of the literature reveals that wasted tires are a threatening waste to the environment. There are several attempts to utilize the wasted tires, but the utilization scale is much less than the wasting rate of tires. In addition, the recycling processes of wasted tires are still expensive and not feasible. Recycling wasted tires for a wealthy application is a novel approach in solar thermal energy storage but is not well studied and characterized.

As the research gap is identified, the current research's objective is to investigate the possibility of utilizing the wasted tires for solar TES. The scope of the work consists of developing an experimental setup and measuring the performance of different TES materials to store solar energy as thermal energy using paraffin wax, pebbles, concrete, and shredded tires. The wasted tires crumbs are investigated as a storage medium or added particles to enhance an existing TES material.

The article is subdivided into the introduction and relevant literature section to identify the research gap. Section 2 outlines the methodology adopted to investigate the effectiveness of the shredded tires as additives to TES materials. Section 3 presents the results gained from the experimental measurements and analysis. Section 4 concludes the findings of the research.

## 2. Materials and Methods

The investigation method is entirely experimental. Accordingly, an experimental setup is designed and fabricated to investigate the TES capacity and compare different compositions of materials with and without added shredded tires. The research procedure is summarized in Figure 3.

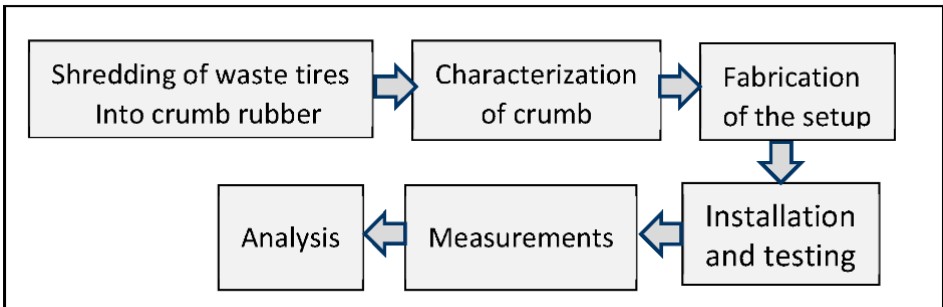

**Figure 3.** Research methodology.

## 2.1. Characterization of Shredded Tires

Generally, crumb rubber is produced by reducing scrap tires to sizes ranging from 3/8″ to 40 mesh particles and removing 99 percent or more of the steel and fabric from the scrap tires. There are several processes for manufacturing crumb rubber. Two of the most common are ambient grinding and cryogenic processing.

A simple procedure has been adopted in the current investigation to characterize the shredded particles. A total of 100 pieces of crumbs were randomly selected and weighted by a digital scale. They are immersed in measured water volume, and the increased volume represents the volume of the 100 particles. The density was obtained by weight over volume. The mean particle volume is 3 mm$^3$, and the mean weight is 0.03 g. According to Liu et al., 2017 crumb rubber is lightweight foamed concrete with a density ranging from 1000 kg/m$^3$ to 1300 kg/m$^3$. Our density from the mean mass per means volume of 100 units of crumb rubber was 1000 kg/m$^3$, which is hypothetically correct.

## 2.2. Experimental Implementations

The experimental setup comprises a 12-mm-thick wood case with 1.23 m length, 1.03 m width, and 0.06 m height. Seven compartments made of 12-mm-thick plywood are located in the wood case. Each compartment is 1 m long, 0.15 m wide, and 0.08 m high. Seven canvas cases were cut to fit inside the compartments. The wood case and the compartments were coated with shellac to reduce the rain and moisture effect on the wood. The entire wood case and compartments were covered by perspex cover to allow solar radiation penetration and avoid the rain. The wood case, cover, and compartments are installed over a metal frame of 0.6 m in height, as shown in Figure 4.

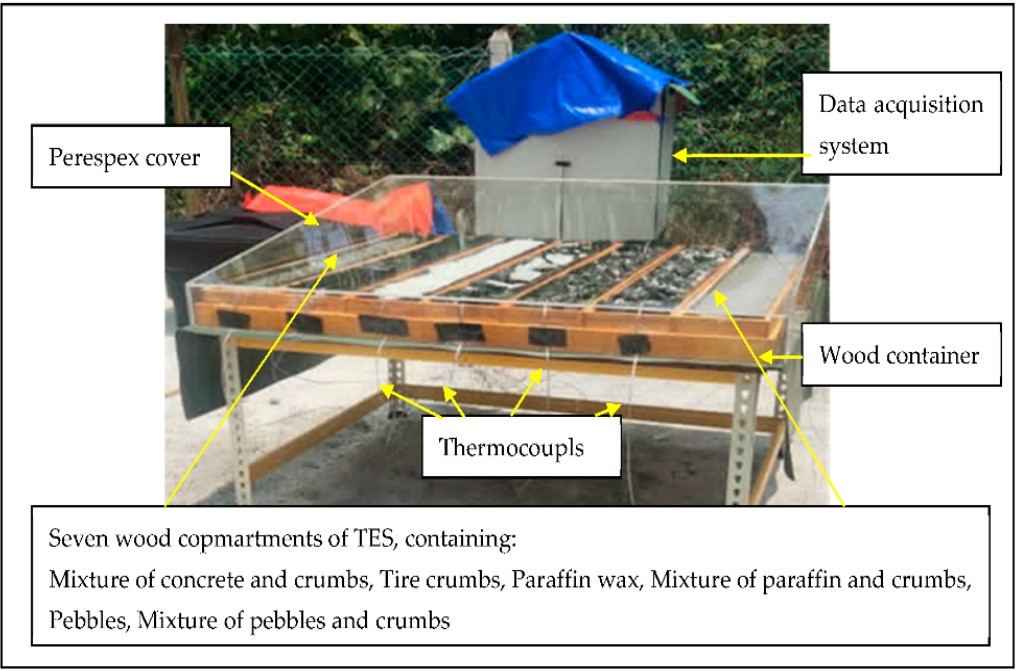

**Figure 4.** The experimental setup located in the solar research site of Universiti Teknologi PETRONAS—Malaysia.

The seven compartments were filled with four base and three composite TES materials, as shown in Table 1. There is no basis for the selection of 40%*vol* shredded tires additive. The 40%*vol* are selected randomly, and further investigations are recommended to evaluate the effect of various percentages of added shredded tires.

**Table 1.** Investigated base and composite TES materials with the compartment coding in the test facility.

| Compartment | TES Material | Type of TES |
|:---:|:---:|:---:|
| C-1 | 100% concrete. (the concrete is 3 parts of sand, 2 parts of cement to 1 part of aggregates) | sensible |
| C-2 | 60%*vol.* Pebbles + 40%*vol.* Shredded Tires | sensible |
| C-3 | 100% Pebbles | sensible |
| C-4 | 60%*vol.* Paraffin Wax + 40%*vol.* Shredded Tires [PCM] | latent |
| C-5 | 100% pure paraffin Wax [PCM] | latent |
| C-6 | 100% Shredded Tires | sensible |
| C-7 | 60%*vol.* Concrete + 40%*vol.* Shredded Tires. (the concrete is 3 parts of sand, 2 parts of cement to 1 part of aggregates) | sensible |

*2.3. Measuring Instruments and Uncertainty Analysis*

The temperatures were measured using a type-J thermocouple, which can measure between −210 to 760 °C with ±0.75% accuracy. All thermocouple wires were connected to a midi data logger, GRAPHTECH GL840, which stores the temperature reading with an accuracy of ±1.55 °C within the temperature range of 0 to 500 °C. The data were then transferred to an excel spreadsheet for data analysis. The measuring device typically used for the incident solar radiation is a pyranometer type MS402 that measures direct and diffuse combined solar radiation within a range of 0 to 4000 $W/m^2$. The pyranometer has an accuracy of ±0.2% nonlinearity and 0.2% tilt response.

Identifying and quantifying the uncertainty in experimental measurement is essential to quantify the expected accuracy. The discrepancy between the *measured value* and the *true value* of a measured variable is expressed in Equation (1).

$$measured\ value = true\ value \pm error \tag{1}$$

The error in the measurement may be sourced from Bias errors and Random errors. Bias is a systematic error due to the instrument's accuracy and resolution. Random error is a non-repeatable inaccuracy caused by uncontrollable events that introduce scattering in the experimental variable values and propagate. Table 2 shows the uncertainty of the measurement variables due to the instruments' accuracy used in the investigation.

**Table 2.** The estimated uncertainties of measurement variables due to systematic errors.

| Variable | Instrument | Accuracy of the Instrument | Maximum Value | Uncertainty and % of Relative Uncertainty |
|---|---|---|---|---|
| Solar Irradiance | Pyranometer, MS402 | ±0.4% due to 0.2% nonlinearity and 0.2% tilt sensitivity | 880 $W/m^2$ | ±3.52 $W/m^2$ or ±0.4% |
| Temperature | Type-J Thermocouples | ±0.75% | 112.5 °C | ±0.84 °C or ±0.74% |
|  | Datalogger | ±1.55 °C up to 500 °C | 112.5 °C | ±0.35 °C or ±0.3% |
| Volume of compartment | Measuring tape | ±3 mm | 0.15 × 0.08 × 1.0 m | $3.6 \times 10^{-5}$ $m^3$ or ±0.3% |
| mass | Digital balance | 0.001 kg | 25.2 kg | 0.0252 kg or ±0.1% |

The ±3 mm accuracy of the volume is due to the 1 mm resolution of tape and the 2 mm propagation of the surface. However, the maximum possible systematic uncertainty in the temperature is ±0.8% due to the accumulation of the thermocouple wire and data logger uncertainties. The maximum uncertainty in the temperature measurement is 2.04% due to

±1.6% random and ±0.8% systematic errors. The resulting maximum uncertainty in the predicted heat gain is ±5.78%, which is acceptable and justifies the accuracy of the results.

The setup is used for the planned investigation for five days of repeatability of measurements. The mean values are selected for the analysis. Five days are sufficient to minimize the uncertainty of the measurements data and the unpredictable weather. The five daily measurements were carried out from 17 March 2021 to 21 March 2021. The temperature was recorded from 9:00 a.m. till 7:00 p.m. every day.

## 3. Results

Evaluation of the storage capacity in the current investigation is rudimentary. The main solar thermal storage parameters are the temperature and the charged/discharged thermal energy by direct solar radiation.

### 3.1. Integral Analysis

Results presented in the form of directly measured temperature values are not enough for proper evaluations and comparisons of the TES compositions. The storage capacity of each compartment is predicted as the amount of energy stored in 30 min using Equation (2).

$$Q_{TES} = m_{TES} \times Cp \times \frac{(T_{i+1} - T_i)}{1800} \tag{2}$$

For the case of the latent TES, using paraffin wax, the total stored energy is some of the gained heat in solid mode, the latent heat, and the gained heat in the liquid phase, as in Equation (3).

$$Q_{TES.\ latent} = m_{TES} \times Cp_{solid} \times \frac{(T_{i+1} - T_i)}{\Delta t} + m_{TES} \times H_l / \Delta + m_{TES} \times Cp_{liq.} \times \frac{(T_{i+1} - T_i)}{\Delta t} \tag{3}$$

The storage capacity (W/kg) of each TES is predicted as the stored energy-to-mass ratio, as in Equation (4).

$$storage\ capacity = Q_{TES} / m_{TES} \tag{4}$$

$Q_{TES}$ (Watt) is the storage material's charged/discharged thermal power. $m_{TES}$ (kg) is the mass of TES material; $Cp$ (J/kg·K) is the thermal capacity of the TES material, and $(T_{i+1} - T_i)$ is the difference between the two consecutive temperature readings.

The properties of each pure material used in the investigation are presented in Table 3. Those values have been used to predict the composite materials' properties and thermal energy storage capacity.

**Table 3.** Properties of the base materials used in the investigations.

| TES Material | Code of the Compartment | Measured Density (kg/m³) | Heat Capacity (J/kg·K) | Mass of TES (kg) |
|---|---|---|---|---|
| Concrete | C-1 | 2400 | 880 | 25.2 |
| Pebbles | C-3 | 1700 | 880 | 17.85 |
| Paraffin | C-5 | 900 | 2500 | 9.45 |
| Shredded tires | C-6 | 1000 | 1230 | 10.5 |

The literature reports the latent heat of fusion of paraffin wax between 200–220 kJ/kg [38]. The measurement carried out in this research showed a 166 kJ/kg value of used paraffin wax.

Rules of mixtures have been used to estimate the properties of the composites that are investigated in the current work.

The density of each mixture, $\rho_m$ is estimated using Equation (5)

$$\rho_m = x_a \times \rho_a + x_b \rho_b \tag{5}$$

The heat capacity of each mixture, $Cp_m$, is estimated using Equation (6)

$$Cp_m = x_a \times Cp_a + x_b Cp_b \tag{6}$$

where $x_a$ is the mass fraction of component $a = \frac{m_a}{m_m}$ and $x_b$ is the mass fraction of component $b, = \frac{m_b}{m_m}$.

The calculations resulted in the quantities shown in Table 4. For all cases of mixtures, component b is the shredded tires with 40%*vol.* or 4.8 kg weight.

**Table 4.** Predicted density and heat capacity of mixtures.

| Mixture | Code of the Compartment | Mass of TES (kg) | Mass Fraction of Component a | Mass Fraction of Component b | Density of Mixture (kg/m³) | Heat Capacity of Mixture (J/kg·K) |
|---|---|---|---|---|---|---|
| 60%*vol.* Paraffin wax + 40%*vol.* shredded tires | C-4 | 11.28 | 0.574 | 0.426 | 942.5 | 1960 |
| 60%*vol.* concrete + 40%*vol.* shredded tires | C-7 | 22.08 | 0.783 | 0.217 | 2096.2 | 956 |
| 60%*vol.* pebbles + 40%*vol.* shredded tires | C-2 | 17.04 | 0.7183 | 0.2817 | 1502.8 | 978.6 |

*3.2. Experimental Environment*

The mean solar irradiation and the ambient temperature of five days of measurement are presented in Figure 5. Results agree with most previous studies and measurements of solar irradiation and ambient temperature in the solar research site. The ambient temperature increased gradually from around 25 °C at 9:00 a.m. to around 35 °C at 2:00 p.m. and then reduced to 27.5 °C at 7:00 p.m. The solar irradiance increases and reaches its highest at around 1:00 p.m., with a value of 880 w/m².

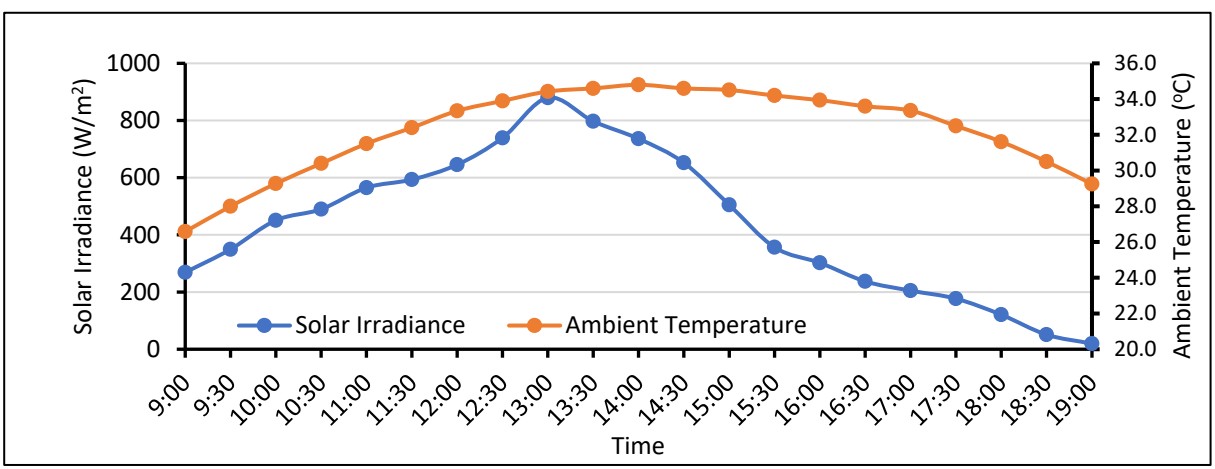

**Figure 5.** The mean values of measured solar irradiation and ambient temperature.

*3.3. Temperature Analysis of TES Materials*

Figure 6 presents the mean of five days of temperature measurements based on 30 min intervals, from 9:00 a.m. to 7:00 p.m. Comparatively, the 100% cement case showed the lowest performance in terms of temperature rise. This low-temperature rise in the cement may be due to the gray color of the top surface and low solar absorptivity. Adding 40%*vol* of shredded tires increased the ability of concrete to absorb higher solar thermal energy. The 100% paraffin wax is indicated.

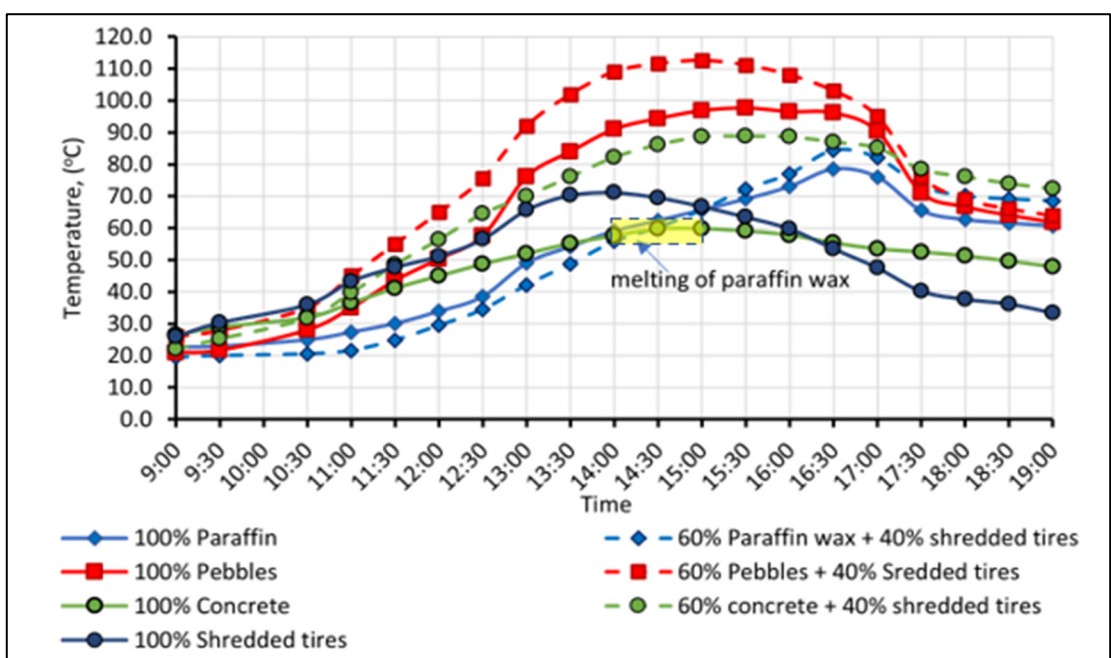

**Figure 6.** Measured temperature as the mean of five-day repeatability of the seven types of TES.

### 3.4. Energy Storage Analaysis

The analysis of the thermal storage is based on the estimated heat absorbed by the material from solar irradiation. Hence, the heat gained and discharged is predicted using equation 1. Figure 7 presents the predicted capacity of the pure and mixed 60%*vol* paraffin and 40%*vol* shredded tires. The pure paraffin's absorption capacity is very small until around 11:00 a.m., then the amount of absorbed and sored heat increases and reaches the maximum at 12.30, where the solar irradiation has a high incidence angle. The added shredded tire crumb slightly increases the storage capacity of the paraffin wax. After 1:00 p.m., both TESs continue storing thermal energy. In both cases, the entire paraffin melts at around 2:30 p.m., and energy storage continues till 4.30 p.m.

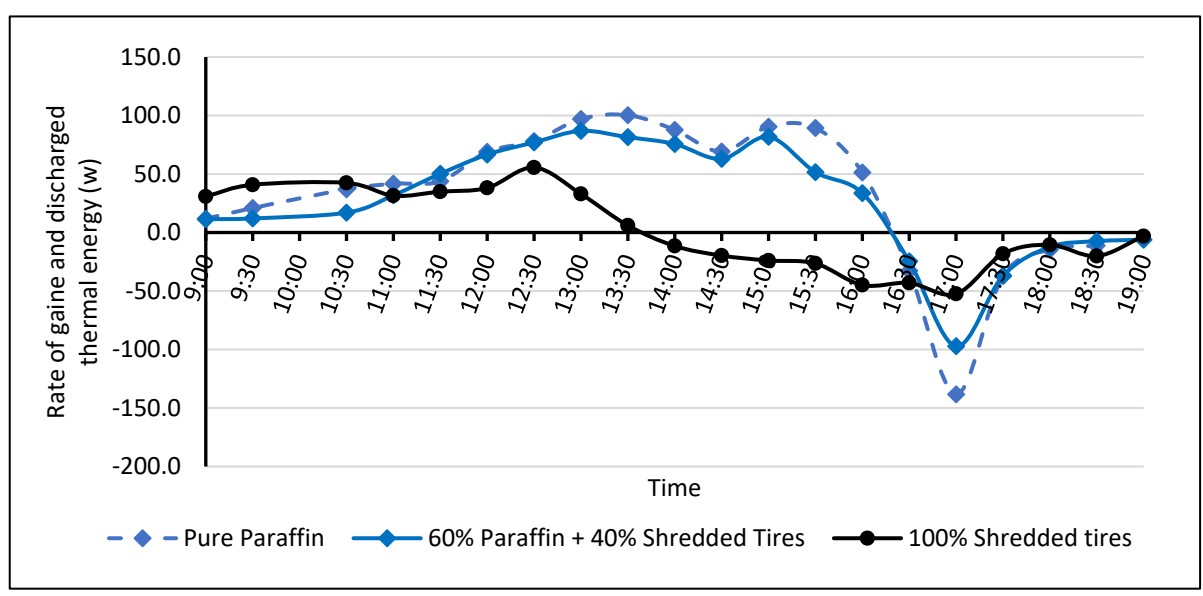

**Figure 7.** Comparison of thermal storage capacity of pure paraffin wax and the composite of 60%*vol.* paraffin wax + 40%*vol.* shredded tires.

Tire crumbs reduce the paraffin wax's capacity to store energy from solar irradiation. Recall the temperature in Figure 5 show that the temperature of the paraffin and paraffin mix started to reduce. As such, the mode of heat energy discharged to the environment starts, and a large amount is disposed to the environment during the half-hour, from 4:00 p.m. to 4:30 p.m. The discharging of heat continues until the end of the measurement period but with a reduced amount. However, the total solar thermal energy absorbed is 888.6 W and 741 W for pure paraffin and paraffin-shredded tire mixture. The mean rate of the TES charging capacity is 127 W/h and 105.8 W/h for the pure and mixed paraffin, respectively.

In the discharging mode, the TES released thermal energy at a mean rate of 80.9 W/h for the pure paraffin wax and 61.4 W/h for the mixture of paraffin wax and crumbs. The added crumbs of shredded tires reduced the capacity of the paraffin wax to charge and discharge thermal energy.

Shredded tires increase the amount of absorbed and stored thermal energy from solar radiation. Figure 8 displays the pure concrete and 60%*vol* concrete + 40%*vol* shredded tire capacity to store and release thermal energy. The thermal energy storage capacity from the solar radiation started from 9:00 a.m. and continued till 2.30 p.m. for the pure concrete and till 3:00 p.m. for the mixture. However, the total amount of solar absorbed and stored thermal energy is 410 W for the pure concrete and 763.6 W for the concrete-shredded tires mixture. There is a 46.3% increase in storage capacity due to added tire crumbs. The added shredded tire also improves the discharge capacity to the concrete. The total discharge is 157.7 W for pure concrete and 208.8 for the concrete mixture.

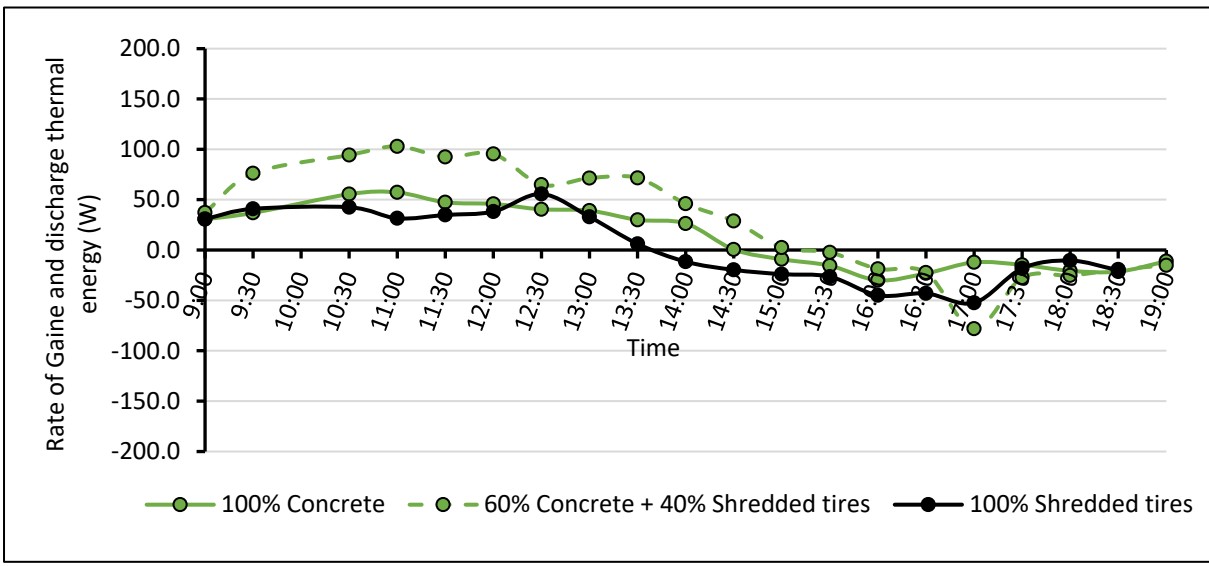

**Figure 8.** Comparison of thermal storage capacity of 100% concrete and the composite of 60%*vol.* concrete + 40%*vol.* shredded tires.

The pebble TES investigation results are presented in Figure 9. The results compare pure pebbles, 60%*vol* pebbles + 40%*vol* shredded tires (in red square symbols), and pure shredded tires crumbs (in black rounded symbols). Added shredded tires to the pebble showed considerable enhancement in the amount of absorbed solar thermal energy. Pebbles and pebbles mixture TES stored thermal energy from 9:00 a.m. till 3:00 p.m. and continued discharging until 7:00 p.m. The absorbed and stored thermal power is 323.6 W by pure shredded tires, 670 W by pebbles, and 850 W by the mixture of pebbles + tire crumbs. The charging period of the shredded tires is shorter than all other tested materials and the discharging period is longer than all others. The discharged thermal energy until 7:00 p.m. is 263 W for the crumbs, 338.6 W for pure pebbles, and 438.4 W for the pebbles and shredded tires.

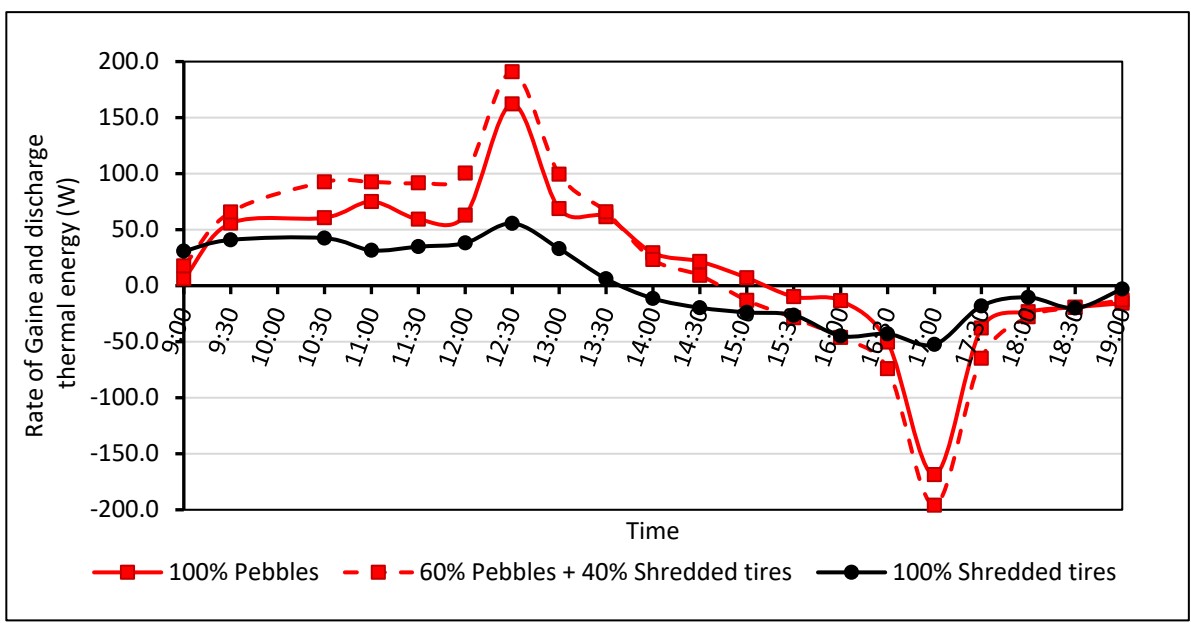

**Figure 9.** Comparison of thermal storage capacity of pure pebbles and a mixture of 60% pebbles + 40% shredded tires.

## 4. Discussion

Selected material and their mixtures with shredded tire crumbs showed different solar thermal energy storage trends during charging and discharging modes. Table 5 summarizes the characteristics and comparison between the seven tested materials and the effect of adding 40%*vol* of shredded tires to the base materials, paraffin wax, concrete, and pebbles during the charging by thermal energy from solar radiation. As base TES materials, paraffin wax shows the highest capacity to store solar thermal than crumbs of tires, concrete, and black painted pebbles.

**Table 5.** The charging behavior of the tested TES materials.

| TES Material and Compartment Code | Charging Period (Hours) | Stored Solar Thermal Energy (W) | Percentage of Enhancement | Rate of Charging (W/h) | Remarks |
|---|---|---|---|---|---|
| 100% shredded tires | 4.5 | 313.6 | - | 69.7 | - |
| Paraffin wax | 7 | 888.6 | | 127 | |
| 60%*vol.* Paraffin + 40%*vol.* shredded tires | 7 | 741 | −20% | 105.8 | Negative effect |
| 100% Concrete | 5.5 | 410 | - | 74.5 | |
| 60%*vol.* Concrete + 40%*vol.* shredded tires | 6 | 783.6 | 42.9% | 130.6 | Positive effect |
| 100% black painted pebbles | 6 | 670 | - | 111.7 | |
| 60%*vol.* black painted pebbles + 40%*vol.* shredded tires | 5.5 | 849.5 | 27.7% | 154.5 | Considerable enhancement |

The predicted solar thermal storage rate values are 127 W/h for paraffin wax, 74.5 W/h for concrete, 111.7 W/h for black painted pebbles, and 69.7 W/h for black painted pebbles shredded tires. Regarding mixtures of 60%*vol* base materials and 40%*vol* shredded tires, the paraffin wax is affected negatively with a 20.8% reduction in the stored solar thermal energy. Furthermore, the rate of charging is reduced from 127.8 to 105.8 W/h. Adding shredded tires to the sensible TES improved the amount of stored thermal energy by 42.9% for the concrete and 27.7% for the pebbles.

Table 6 displays the discharging characteristics of the seven tested materials. Furthermore, it compares the effect of adding 40%*vol* of shredded tires to the base materials during the thermal energy discharging mode. Among the base TES tested materials, the largest amount of heat released is by the pebble with 338.6 W. The other materials, shredded tires, paraffin wax, and concrete, discharged thermal power of 273.4, 242.6, and 157.7 W, respectively. The discharged amounts of stored thermal power from the mixtures of 60%*vol* paraffin, 60%*vol* concrete, and 60%*vol* pebbles with 40%*vol* of tire crumbs are 184.3, 208.8, and 483.4 W, respectively.

**Table 6.** Discharging behaviors of the tested TES materials with and without 40%*vol* added shredded tires.

| TES Material and Compartment Code | Discharge Period (Hours) | Released Thermal Energy (W) | % Enhancement (%) | Rate of Discharging (W/h) | Remarks |
|---|---|---|---|---|---|
| 100% shredded tires | 5.5 | 273.4 | - | 49.7 | - |
| Paraffin wax | 3 | 242.6 | - | 80.9 | - |
| 60%*vol.* Paraffin + 40%*vol.* shredded tires | 3 | 184.3 | −31.6 | 61.4 | Crumbs reduced the discharge capacity |
| 100% Concrete | 4.5 | 157.7 | - | 35 | - |
| 60%*vol.* Concrete + 40%*vol.* shredded tires | 4 | 208.8 | 24.4 | 52.2 | Small enhancement |
| 100% Black painted pebbles | 4 | 338.6 | - | 84.7 | - |
| 60%*vol.* Black painted pebbles + 40%*vol.* shredded tires | 4.5 | 483.4 | 30% | 107.4 | Considerable enhancement |

The added shredded tires reduced the amount of discharged heat by the latent type, paraffin wax, by 31.6%. However, adding 40%*vol* of tire crumbs to the sensible TES, concrete, and pebbles has enhanced the discharged amount of stored thermal energy by 24.4% and 30%, respectively.

The reduced performance of the paraffin wax is attributed to the tire crumbs' slow charging and discharging of the solar thermal energy. The presence of the crumbs in the paraffin wax causes absorption of the thermal energy by the crumbs. It alters the melting of the solid paraffin wax and heat storage in the melted paraffin. Most of the energy converted from solar radiation and absorbed as thermal energy by the paraffin is utilized for the phase change. After melting, the liquid phase temperature of paraffin increases, as shown in Figure 5, while the temperatures of sensible materials are reduced. Paraffin wax was noted to start melting at around 58 °C.

The shredded tire crumbs in the sensible materials, concrete, and pebbles enhanced the absorptivity of solar energy. In the case of the concrete, the black color of the tire crumbs increases the solar absorption compared to the gray color of the concrete. Testing this fact by further investigating the concrete with and without tires crumbs, with and without black surface painting is recommended.

In the case of a large enhancement in pebbles' storage capacity, the shredded tires' size is small enough to fill the gaps between the pebbles. Instead of air-filled gaps between the pebbles, the solid particles increase the surface contact between the pebbles and between the pebbles and crumbs. The tire crumbs penetrate the pebbles and reduce the porosity, leading to the high performance of photothermic solar energy conversion at the bed surface and increasing the pebbles' thermal energy absorptivity and storage. Figure 10 compares the charging and discharging capacity of the tested types of TES materials.

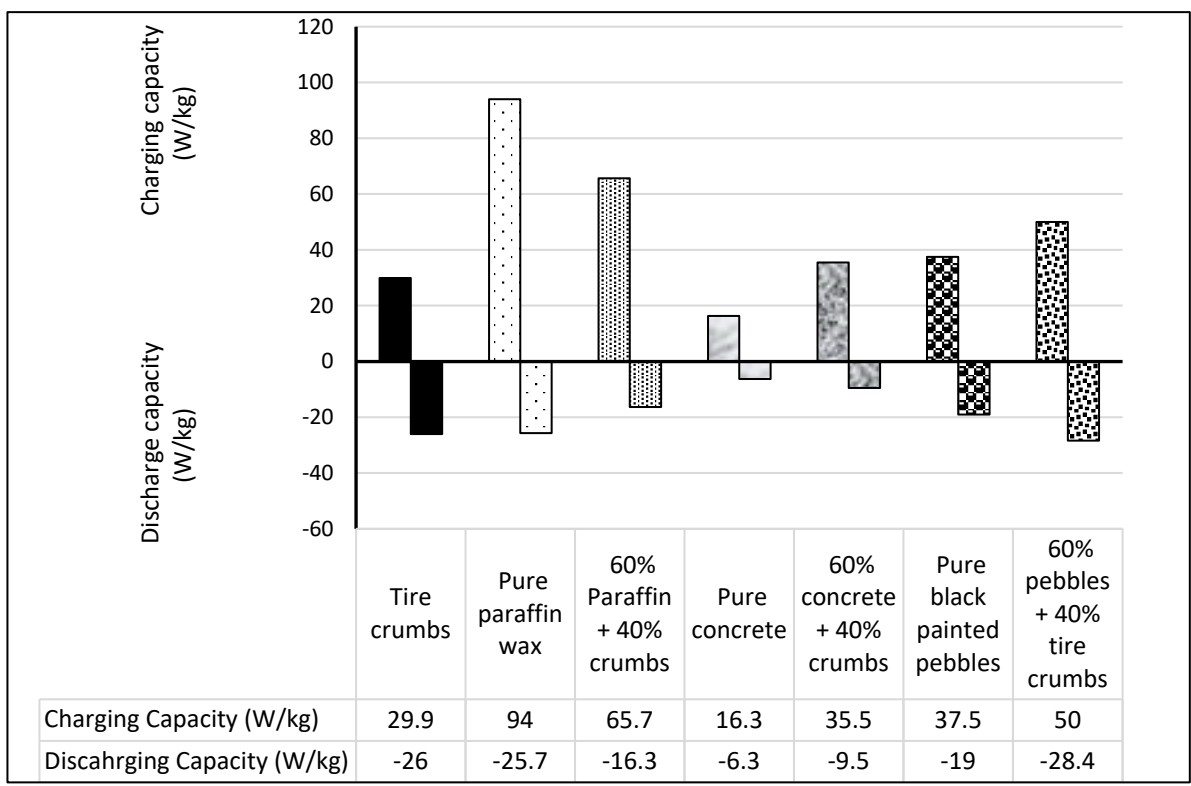

**Figure 10.** Comparison of the charging and discharging capacity of the seven tested TES materials.

The contribution of the added tire crumbs to the tested base TES materials is summarized in Table 7. Adding tire crumbs to the paraffin wax has a negative effect, where the charging capacity is reduced by 42.9%, and the discharging capacity is reduced by 57.7%. In contrast, the concrete and the pebbles show enhanced storage capacity. Adding 40%*vol* of crumbs to the concrete increased the charging capacity by 54% and discharging capacity by 33.7%. The 40%*vol* added tire crumbs to the pebbles increased its charging capacity by 25% and the discharging capacity by 33%.

**Table 7.** Charging and discharging capacities of the tested materials and the effect of adding tire crumbs on the base materials storage capacities.

| TES Material and Compartment Code | Charging Capacity (W/kg) | Percentage of Enhancement (%) | Discharging Capacity (W/kg) | Percentage of Enhancement (%) | Remarks |
|---|---|---|---|---|---|
| 100% shredded tires | 29.9 | | 26 | | - |
| Paraffin wax | 94 | | 25.7 | | - |
| 60% Paraffin + 40% shredded tires | 65.7 | −43 | 16.3 | −57.7 | Crumbs reduced the capacity |
| 100% Concrete | 16.3 | | 6.3 | | - |
| 60% Concrete + 40% shredded tires | 35.5 | 54 | 9.5 | 33.7 | Large enhancement |
| 100% Black painted pebbles | 37.5 | | 19 | | - |
| 60% black painted pebbles + 40% shredded tires | 50 | 25 | 28.4 | 33 | Good enhancement |

*The contribution* of the current research is represented in the proof that wasted tires can be used in energy systems. The analysis of measurement results demonstrates that solar absorptivity and photothermic conversion by the sensible types of TES materials, such as pebbles and concrete, are improved by adding tire crumbs. However, the storage amount of latent type of TES material, such as paraffin wax tested in this work, is reduced by adding tires crumbs.

*The limitation* of the work could be identified in the limited types of tested TES base materials, paraffin wax, concrete, and pebbles. Furthermore, only one amount of 40%*vol* added shredded tires to the base TES materials. Further mixing value is recommended to be investigated.

Moreover, it is recommended to test the TES as a solar water heating medium by passing water through pipes embedded in the TES and evaluating the solar water heating performance.

## 5. Conclusions

Shredded wasted tires crumbs are assessed for the possibility of utilization in the energy application. Concrete, paraffin wax, and pebbles are selected as solar thermal energy storage materials, and their enhancement by adding 40%*vol* tire crumbs is investigated experimentally. The 40%*vol* added tire crumbs to 60%*vol* base TES materials showed a negative effect on the thermal storage capacity of paraffin wax, which is latent TES material. However, they increased the thermal storage capacity of concrete and pebbles, which are sensible TES materials. The added tire crumbs reduced paraffin wax's storage capacity (W/kg) by 43%. In comparison, it increased black painted pebbles and concrete storage capacities (W/kg) by 54% and 25%, respectively. Added shredded tires improved the performance of sensible TES but reduced the performance of latent TES.

A mixture of shredded tires and pebbles is a promising method to enhance the solar updraft system's performance as it shows considerably enhanced solar thermal storage capacity and high temperature. Furthermore, it is recommended to investigate solar water heating utilizing the tested materials for direct solar absorptivity and heating water in pipes flow.

The results reveal the probability of a circular economy using wasted tires with sensible TES for solar-to-thermal energy conversion.

**Author Contributions:** Conceptualization, H.H.A.-K. and P.R.; methodology, H.H.A.-K. and B.A.B.; formal analysis, H.H.A.-K. and P.R.; investigation, B.A.B. and P.R.; writing—original draft, P.R.; writing—review, H.H.A.-K. and E.M.; validation, E.M.; supervision, H.H.A.-K.; project administration, H.H.A.-K. and B.A.B.; funding acquisition, H.H.A.-K. All authors have read and agreed to the published version of the manuscript.

**Funding:** This research was funded by Universiti Teknologi PETRONAS under the RIPHEN project, grant number 015MD0-063 on the Sustainable Smart Buildings.

**Acknowledgments:** The authors acknowledge Universiti Teknologi PETRONAS for the financial and logistic support to conduct the research at the solar research site.

**Conflicts of Interest:** The authors declare no conflict of interest. The funders had no role in the study's design, in the collection, analyses, or interpretation of data, in the writing of the manuscript, or in the decision to publish the results.

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
