# Peer review of "Rudimentary Assessment of Waste-to-Wealth of Used Tires Crumbs in Thermal Energy Storage"

_recycling, doi:10.3390/recycling7030040_

Round 1
Reviewer 1 Report
The authors investigated the possibility of using shredded waste tires as sensibleThermal Energy Storage (TES) materials. The findings support utilization of wasted tires with sensible TES for solar-to-thermal energy conversion, favoring a circular economy. The research is practical and useful and the topic falls within the scope of the journal. The following comments are proposed to further improve the quality of the manuscript before acceptance.
Point 1: Authors must look into the results section. Discuss the key findings.
Point 2: Extensive English editing is required.
Point 3: The structure of the paper is not sound.
Point 4: Authors must try to mention the contribution of the study in last of Abstract and Conclusion. It is one of the critical conditions.
Point 5: The abbreviation “TES” should not only be defined in the Abstract but also in the main test.
Point 6: There are only 10 publications in the reference list, which is too insufficient for a regular research paper. Some other useful references for this work should be considered and discussed: Construction and Building Materials 48 (2013): 636-646; Construction and Building Materials 299 (2021): 123939; Nano Energy 89 (2021): 106376.
Author Response
The authors investigated the possibility of using shredded waste tires as sensible Thermal Energy Storage (TES) materials. The findings support utilization of wasted tires with sensible TES for solar-to-thermal energy conversion, favoring a circular economy. The research is practical and useful and the topic falls within the scope of the journal. The following comments are proposed to further improve the quality of the manuscript before acceptance.
Point 1: Authors must look into the results section. Discuss the key findings.
Authors' response: As suggested, the results section has been expanded, and more results tables are added, with insightful analysis and discussion. Figure 9 has been modified for moor with discussion and analysis of results.
Point 2: Extensive English editing is required.
Authors' response: as recommended, the article is subjected to proofreading and English commands are improved.
Point 3: The structure of the paper is not sound.
Authors' response: the conventional procedure to write a research article is followed. We started with an introduction and literature related to the subject matter, then we described the investigation method, then we presented the results, then we discussed the findings and ended the article with conclusions.
Point 4: Authors must try to mention the contribution of the study in last of Abstract and Conclusion. It is one of the critical conditions.
Authors’ response: correct. The comment is valid and constructive. We have added statements as suggested. We emphasized the importance of the work to the circular economy.
Point 5: The abbreviation “TES” should not only be defined in the Abstract but also in the main test.
Authors' response. Thank you for the note. We defined TES in the first appearance in the introduction, section 1.6.
Point 6: There are only 10 publications in the reference list, which is too insufficient for a regular research paper. Some other useful references for this work should be considered and discussed: Construction and Building Materials 48 (2013): 636-646; Construction and Building Materials 299 (2021): 123939; Nano Energy 89 (2021): 106376.
Authors response. Thank you for the advice. The number of references is increased from 10 to 38.

Reviewer 2 Report
REVIEW:
Reviewer's Assumptions:
The topic is about modeling, it aims to assess the possible use of shredded used tires in thermal energy storage (TES) applications, both in sensitive and hidden materials. Modeling is based on thermo-mechanical models of technical conditions, postulated states. Models should include, on the one hand, the postulated states of the power industry target outputs: product quality (QoP) of power, efficiency of the thermal process (EoPc), harmlessness of the product and process (HoPaPc). On the other - inputs and accompanying relations: technical conditions (TC) including control levels (AI); prices of carriers, energy sources (PoS), power and energy prices (PoPaE), emission and harmfulness costs (CoE) and permissible variability of everything over time (VAoT), etc.
Adequate knowledge:
Research modelling towards condition assessment is a structured and knowledge-based procedure. Ordering includes: mathematical model of the research object (function of the research object); conditions, permissible area (restrictions); criteria and research (theoretical, experimental). The article presents adequate, multifaceted, partially coordinated and susceptible to decisions and control knowledge (mathematical models). The state of knowledge and practice of the issue were based on 10 bibliographic items. The content of the article, only partially provides a comprehensive overview of knowledge.
Research skills and instruments:
Instrumentation includes target output tracking: black carbon in used tires can harvest solar energy in several ways. Soot is mixed and pressed with exfoliated graphite. This mixture was pressed onto a small copper disc in a preliminary investigation. The disk was connected to an insulated body of water using pipes that generated a small solar geyser. At its peak, the water temperature rose to 45 °C, about 4 kW.h/m2 of solar energy was obtained. With an average irradiance of about 6 kW.h/m2, an efficiency of about 75% was achieved.
Purposefulness and creative attitude:
An example of deliberate, mechanical support for innovative transformations and states of quality, efficiency, harmlessness of energy depends to a very large extent on creative and research capabilities. The scope of creativity and research consists in developing an experimental configuration and measuring the performance of various TES materials for storing solar energy as thermal energy using paraffin wax, pebbles, concrete and shredded tires. Tire chips are studied as a storage medium or added particles to improve the existing TES material.
The introduction and the state of the literature are included; the methodology adopted to investigate the effectiveness of shredded tyres as additives to TES materials is presented; present the results and analyse the results; with a summary. It was emphasized that the accuracy of heat management models in the mix of accumulating materials is seriously affected by the uncertainty of various factors, such as the thermo-mechanics of porous media and insulators.
Conclusion:
Critical remarks: I have no critical remarks, I only lacked a formal formulation of the goal and a general problem leading to the achievement of the modeling goal, for example: What is the scientific goal, and what is the practical purpose of the study? What is the problem that has been formulated to achieve the scientific goal, and what problem has been formulated to achieve the practical goal? In the future, it is worth thinking and designing hard assumptions, conditions and modeling goals for optimization.
And finally: I assess the work positively!
Author Response
Responses to reviewers' comments
Before starting detailed responses, we are marking our gratitude to the reviewer for the time and effort to read and address the weaknesses in the article. Your valuable comments and constructive suggestion have been considered in the revised version.
We hope that our amendments satisfy you.
Reviewer 2
Reviewer's Assumptions:
The topic is about modeling, it aims to assess the possible use of shredded used tires in thermal energy storage (TES) applications, both in sensitive and hidden materials. Modeling is based on thermo-mechanical models of technical conditions, postulated states. Models should include, on the one hand, the postulated states of the power industry target outputs: product quality (QoP) of power, efficiency of the thermal process (EoPc), harmlessness of the product and process (HoPaPc). On the other - inputs and accompanying relations: technical conditions (TC) including control levels (AI); prices of carriers, energy sources (PoS), power and energy prices (PoPaE), emission and harmfulness costs (CoE) and permissible variability of everything over time (VAoT), etc.
Authors’ response. With all the respect due, the work is on an experimental assessment of the influence of adding tire crumbs to TES materials and evaluating the storage capacity. We don’t have a model to predict the product quality: (QoP) of power, the efficiency of the thermal process (EoPc), or the harmlessness of the product and process (HoPaPc).
On the other issue, inputs and accompanying relations: technical conditions (TC), including control levels (AI); prices of carriers, energy sources (PoS), power and energy prices (PoPaE), emission and harmfulness costs (CoE) and permissible variability of everything over time (VAoT), etc all those are out of the scope of the work.
The theme is to evaluate the thermal performance of TES with and without added tire crumbs. The suggested parameters by the respected reviewer could be a continuation of the research and can be carried out independently.
Adequate knowledge:
Research modelling towards condition assessment is a structured and knowledge-based procedure. Ordering includes: mathematical model of the research object (function of the research object); conditions, permissible area (restrictions); criteria and research (theoretical, experimental). The article presents adequate, multifaceted, partially coordinated and susceptible to decisions and control knowledge (mathematical models). The state of knowledge and practice of the issue were based on 10 bibliographic items. The content of the article, only partially provides a comprehensive overview of knowledge.
Authors’ response. Thank you for the comment. The number of references to support state-of-the-art in the field is increased from 10 to 38. Hence, comprehensive knowledge, as the best possible, is stated in the article. However, we are not modelling the process. We are experimenting and measuring the phenomena utilizing an experimental setup.
Research skills and instruments:
Instrumentation includes target output tracking: black carbon in used tires can harvest solar energy in several ways. Soot is mixed and pressed with exfoliated graphite. This mixture was pressed onto a small copper disc in a preliminary investigation. The disk was connected to an insulated body of water using pipes that generated a small solar geyser. At its peak, the water temperature rose to 45 °C, about 4 kW.h/m2 of solar energy was obtained. With an average irradiance of about 6 kW.h/m2, an efficiency of about 75% was achieved.
Authors' response. Thank you for sharing the interesting information. However, this article deals with tire crumbs and mixtures of tire crumbs with TES materials. We are not using powder. The intention is to produce cheap and affordable energy materials. In the authors' opinions, the assessment reached its aim and evaluated the effect of tire crumbs on TES performance.
Purposefulness and creative attitude:
An example of deliberate, mechanical support for innovative transformations and states of quality, efficiency, harmlessness of energy depends to a very large extent on creative and research capabilities. The scope of creativity and research consists in developing an experimental configuration and measuring the performance of various TES materials for storing solar energy as thermal energy using paraffin wax, pebbles, concrete and shredded tires. Tire chips are studied as a storage medium or added particles to improve the existing TES material.
The introduction and the state of the literature are included; the methodology adopted to investigate the effectiveness of shredded tyres as additives to TES materials is presented; present the results and analyse the results; with a summary. It was emphasized that the accuracy of heat management models in the mix of accumulating materials is seriously affected by the uncertainty of various factors, such as the thermo-mechanics of porous media and insulators.
Authors' response. Thank you for the insight into our work. As it is rudimentary, we have opened a line for further studies on using tire crumbs. Further studies can investigate the effect of different added percentages of tire crumbs. Also, studying more types of latent and sensible TES materials with added tire crumbs is recommended.
Conclusion:
Critical remarks: I have no critical remarks, I only lacked a formal formulation of the goal and a general problem leading to the achievement of the modeling goal, for example: What is the scientific goal, and what is the practical purpose of the study? What is the problem that has been formulated to achieve the scientific goal, and what problem has been formulated to achieve the practical goal? In the future, it is worth thinking and designing hard assumptions, conditions and modelling goals for optimization.
Authors' response. We have considered the respected reviewer's remarks and addressed them in the conclusions section, one by one. Thank you for the valuable and constructive remarks.
And finally: I assess the work positively!
Authors' response. We appreciate your interest, comment and support.

Reviewer 3 Report
Please refer to the attached file

Author Response
Reviewer 3 Comments
Title: Rudimentary assessment of waste-to-wealth of used tires crumbs in thermal
energy storage (recycling-1731623)
Decision: Major revision
Technical Comments
- In ‘Abstract’, Please re-write the abstract following the pattern [background] -> [objective] -> [process] -> [results] -> [contribution]
Authors’ response. Thank you for the advice. We followed the suggestion and revised the abstract to show a background and problem statement, the objective or target of the study, the process or investigation process. We showed the main findings from the results and the contribution of the work to the circular economy.
- In Introduction, the literature review was a lack of contents. I recommend that the author need to add the literature review to provide a solid basis for the study.
Authors' response. The respected reviewer is correct. We adopted the criticism and expanded the introduction, including a literature review on the subject matter by increasing the citations from 10 to 38.
- The research process is very simple, so it is hard to understanding. I recommend that research framework figure need to be attached to this paper in detail.
Authors' response. The results opened a door for further investigations on the use of tire crumbs for TES and solar heating. The research process methodology is simple as the work is a rudimentary assessment. The framework or the research flow process is presented in figure 3. Figure 1 is revised to labling the experimental setup and make it more understandable.
- The authors conducted the experimental implementation about 7 compartment. Why the authors selected the 7 compartments? I think the authors have a criteria for selection with respect to 7 compartments. So, The selection of criteria should be explained in detail. If the author selected the compartment according to the subjectivity, I would disagree the experimental plan.
Authors’ response. The selection criteria are based on what we use in the other TES research site. In many projects on our research site, we have Solar Chimney and Solar vortex engine integrated with pebbles TES. In other projects, we used the paraffin wax as a base material for TES integrated with a solar water heater. We used the concrete as TES integrated with a solar drying system in another project. Accordingly, we designed the system to test seven TES materials. Those three TES materials + three mixtures + pure tire crumbs make it seven types. Further materials with various added percentages of tire crumbs can be investigated in future studies.
- The author wrote that the experimental implementation was conducted for five days. Then, the author should analyze the descriptive statistics in results and discussion.
Authors response. Good suggestion. We have added a section for the uncertainty analysis in the methodology section.
- In Figs 5-7, I recommend the error bar should be inserted by each figures based on collected data.
Authors' response. We have followed the normal procedure of presenting solar system results over the day. The usual practice is to present a curvilinear pattern of the variable with time. However, we added one more figure in the histogram “bar” type to meet the reviewer's idea. Also, we added three more tables to support the analysis of the results.
- I think the solar irritation and ambient temperature are maximized from 13:00 to17:00. So, to enhance the understanding to the readers, the authors present the weather information not irritation and temperature but others at the time of the experiment as a graph and explain the research results considering weather condition at the time of the experiment.
Authors' response. Agree. It is common to show the experimental environment in terms of ambient temperature and solar irradiance. Since we have all the records of the varials, we added a new figure showing the mean of the five days of measurements. see section 3.2., please.
- The research contribution should be clearly stated and more elaboration on the practical and theoretical implications of the findings from this study should be provided.
In Conclusion, the author presents the research contribution, limitation and future work. However, It is not clear. The author should be explained in detailed and clear.
Authors' response. Thank you for addressing the article's setback due to a research contribution shortage. As mentioned in your conclusion comment, we have addressed the contribution. We have considered the comment in the revised version and emphasized the research contribution and practical implications.
Also, we agree with the reviewer that the limitations should be highlighted to allow other researchers to improve in the research line of waste tire utilization for energy applications. We added text to meet the reviewer's comments at the end of the conclusions in terms of a recommendation for future works.
See the last added Pars before the conclusions, please.

Round 2
Reviewer 1 Report
Although the author made some changes, the structure, result presentation, references and especially the innovative embodiment are still far from a paper that can be published. The authors should reconsider the suggestions proposed in the first round and should provide a much better revisions in the next round. Especially the authors must try to mention the contribution of the study in last of Abstract and Conclusion and the reference list need a thorough improvement. They are the critical conditions for acceptance for publication.
Reviewer 3 Report
The authors addressed all my comments.